# Understanding Leadership from the Inside: Using Ethnographic Methods to Examine How the Interplay between Leaders, Followers, and Group Context Shapes Leadership Outcomes

**DOI:** 10.3390/bs14100946

**Published:** 2024-10-14

**Authors:** James Coleman, Clifford J. Mallett, Niklas K. Steffens, S. Alexander Haslam

**Affiliations:** 1School of Human Movement and Nutrition Sciences, University of Queensland, St. Lucia, QLD 4072, Australia; cmallett@uq.edu.au; 2School of Psychology, University of Queensland, St. Lucia, QLD 4072, Australia; n.steffens@uq.edu.au (N.K.S.); a.haslam@uq.edu.au (S.A.H.)

**Keywords:** leadership, ethnography, leadership development, identity leadership

## Abstract

This paper outlines a novel method for leadership researchers and practitioners to understand how and why effective and ineffective leadership look different in different groups. Leadership is a complex and contextually dependent process influenced by the interplay between leaders, followers, the group, and their environment. The social identity approach to leadership describes how a group’s identity shapes the ways in which people can lead effectively. It also implies that (in)effective leadership looks different across diverse groups and teams. Accordingly, it follows that there is no single correct way to lead. To explore these ideas, we propose ethnographic methods, where researchers and practitioners immerse themselves in a group environment, as a novel type of method for examining leadership in action. We suggest the social identity approach as a framework to help guide researchers’ data collection and sense-making of leadership behaviours. Additionally, we explain that ethnographic data can be represented well through creative non-fiction stories that capture the context surrounding leadership behaviours. These stories could support leadership consultancy and development programs to demonstrate the complex interplay between leaders, followers, and the group context.

## 1. Introduction

Effective leaders propel their teams towards the achievement of collective goals. Coaches, athletes, and management staff who lead effectively contribute to their team’s success [1]. In sports, effective leadership has been linked to improved cohesion, confidence, performance, enjoyment, motivation, effort, and well-being [2,3,4,5,6]. Given the critical role of leadership in team functioning, researchers have long sought to identify effective leadership behaviours and ways to develop those behaviours in leaders.

While many positive outcomes have been attributed to effective leadership, the pathways to achieving these outcomes are diverse. For instance, effective leadership can sometimes be autocratic, with decision-making power exercised mainly by a leader, while in other cases, it can be democratic and revolve around group decision-making [7]. Other approaches encourage leaders to build the internal motivation of team members [8] or to transform team members’ morals and professionalism through transformational and ethical leadership [9,10]. Effective leadership is therefore not a matter of discerning the ‘best’ leadership approach but rather a matter of identifying helpful leadership behaviour for a specific group and its context. It follows too that effective leadership in high-performance environments requires leaders to be flexible in their approach to leadership. In this sense, leaders are encouraged to be like chameleons who adapt their leadership style to meet the demands of the group they are leading in a given situation [11].

Leadership occurs in group environments. For instance, a leader is always a leader *of* something like an organisation, a team, or a nation. However, formal appointments of individuals to positions of power (e.g., a CEO, manager, coach, captain, parliamentarian, or head of state) do not limit the potential for other group members to lead. Leadership is about motivating and inspiring fellow group members to *want* to do things for the group’s benefit, not getting them to comply [12]. Consider a colleague or classmate who supports, challenges, and inspires their peers to produce higher quality work, a teammate who consoles a player they notice is struggling, or community members who round up friends to support neighbours as they prepare for an incoming flood. All group members have the potential to display leadership, not just those who are in formal leadership positions [12,13]. The yardstick of leadership is not a person’s title or office, but the influence they have and are seen to have by others, and those others’ perceptions of their dependability.

Leadership is a complex interpersonal process influenced by social, cultural, political, and physical environments [12]. One of the challenges of developing effective leadership programs is that the context shapes the conditions for effective leadership. Alvesson [14] argues that leadership research has often provided simplified, universal, and idealised conclusions that overlook the context in which leadership occurs. Such an approach accepts that wherever a leader works—for example, on a construction site, in an accounting firm, in the army, in a kindergarten, or in a professional sports team—the same leadership qualities and behaviours will apply. It has also been observed that leadership research has typically focused on formal leaders in positions of power (e.g., a CEO, manager, captain, or coach), and has overlooked the agency of other group members as informal leaders and followers with unique values, norms, and behavioural preferences [14,15,16]. Yet without group members following, there is no leadership [15,16]. A comprehensive understanding of leadership must therefore consider the followers and the surrounding context in which leadership occurs.

To understand how the context shapes leadership, research needs to be situated within the specific environment in which leadership occurs [14]. To this end, we propose using ethnographic methods, in which researchers immerse themselves within a group or team, as an important yet underutilised means to better understand leadership. We suggest that by immersing themselves for an extended period, researchers can gain deeper insights into a group or team’s environment and dynamics. In what follows, we describe the social identity approach to leadership as a helpful framework for understanding the relationship between leaders and followers. We then go on to discuss how researchers and practitioners can immerse themselves in group and team environments to understand the context in which leadership occurs, and conclude by outlining how researchers might apply their findings to leadership consultancy and development programs.

## 2. The Social Identity Approach

As noted above, leadership analyses typically emphasise the importance of the leader and their actions and characteristics for effective leadership. For example, it has been suggested that leaders should generally possess traits such as charisma and intelligence [17,18]. However, as Haslam and colleagues [12] highlight, the meaning of these leadership qualities varies according to the context and perceptions of the followers. For example, a highly successful football coach might be seen as intelligent and charismatic within a football team, yet in a chess tournament, that same coach is unlikely to be seen as either. One reason for this is that qualities such as charisma and intelligence are *attributed* to leaders by followers on the basis of perceived shared group identity [12,19]. In a football team, the team is more likely to perceive the successful football coach as intelligent and charismatic because of a shared ‘footballer’ identity that values football knowledge. But in a chess tournament, that same coach would not share an identity with the chess players and would not be perceived as ‘one of us’, and so, as a result, the players are less likely to come to see that leader as intelligent or special in some other way. The point here is that leadership always inheres in a particular group (e.g., a club, a sport, a profession), and it is this group context that structures the relationships between leaders and followers [12]. Therefore, a comprehensive leadership analysis should consider the group around which leaders and followers cohere and the context in which they find themselves.

The social identity approach to leadership provides an avenue for understanding the leader–follower relationship by starting with the assertion that people see themselves not only as unique individuals (as ‘I’ and ‘you’ in terms of personal identity) but also as group members (as ‘we’ and ‘us’ in terms of social identity) [20,21]. When people recognise each other as part of the same ‘us’, they can coordinate their efforts and work together towards their group’s goals [20,21]. To provide a very basic example, in a football game, distinguishing between ‘us’ and ‘them’ is crucial to knowing who to pass the ball to and who to tackle [12]. This shared sense of social identity makes the football game work. Indeed, without social identities, the football game might look more like a flock of seagulls chasing hot chips.

A shared social identity is established not only on the basis of the characteristics that group members have in common but also with reference to the unique qualities that distinguish one group from other similar groups. In short, it is about what makes ‘us’ special. For example, one organisation may view themselves as ‘creative’ and ‘playful’ compared to another organisation that views themselves as ‘methodical’ and ‘structured’. For the employees in the first team, working in a creative and playful way can establish and strengthen a shared social identity. Whereas, if this team were managed to function like the second organisation, those employees might feel misunderstood and coerced into conforming to a style of work that contradicts what makes the organisation unique and special. As a shared social identity becomes woven into the fabric of group members’ sense of self, they come to realise that their goals and aspirations are shared—that ‘we’ are in this together. They understand that working together is essential for success. And this shared sense of ‘us’ is then the basis for cooperation and trust within the group [12,22].

Effective leaders are perceived by their followers as part of ‘us’ and to be doing it for ‘us’ [12]. People will follow a leader who represents the group’s ideal qualities and who is believed to have the group’s best interests at heart. On the other hand, regardless of a leader’s actions, people are unlikely to follow a leader who is perceived to be one of ‘them’ or to be placing their own self-interest above the group’s interests [23]. This leader–follower relationship highlights the danger of focusing exclusively on leadership behaviours—for how can we determine whether a leader’s behaviour is effective or ineffective without understanding how a particular group of followers perceive their leader?

Effective leadership also involves creating and cultivating a sense of ‘us’ among the group that is being led. This requires leaders to reflect on the content of a group’s identity, such as its shared values, norms, goals, and preferences for behaviours. This is important in order for a potential leader to be in a position to enact and represent what ‘we’ are about. Finally, leaders also need to realise the group’s shared goals and ambitions by bringing these qualities to life in a way that directs the group towards their shared goals. Hence, the essence of effective leadership is embedded and inseparable from the group itself [12].

Leadership programs that are informed by the social identity approach can help leaders and followers collaboratively uncover their unique qualities and work more effectively as a team [24,25]. However, to enhance their usefulness and minimise the chances of potential backlash, it is important to understand the contextual nuances that influence leadership behaviours. For example, meta-analytic data support the use of on-site (rather than off-site) leadership training [26]. This is because off-site programs often follow a one-size-fits-all approach that fails to capture the context and needs of group members and their organisation. More impactful leadership development is tailored to the context and needs of the group [26].

The social identity approach provides a framework for leaders and researchers to understand how relevant contextual factors such as group identities influence leadership. Leaders and researchers who understand the qualities of a group’s identity are better placed to develop leadership strategies and enact behaviours tailored to the group. These behaviours and strategies are most effective when a leader who enacts these behaviours is perceived by their followers as one of ‘us’ and to be doing it for ‘us’. In isolation, examining a leader and their behaviour provides just one piece of the puzzle. For a more complete picture of leadership, we need to consider the relationship between leaders, followers, and the context.

There is of course a wide range of other leadership approaches that are pursued across diverse settings, including those of sport and exercise, education and learning, and business and the workplace. However, as intimated above, many of these other approaches focus primarily on the individual leader and overlook the ways in which the interplay between leaders, followers, the group, and the context influences leadership [14]. Researchers and practitioners who consider the unique qualities of the group and the context in which leadership occurs can obtain a rich understanding of leadership. Unlike other leadership frameworks, the social identity approach to leadership provides an avenue for understanding the relationship between leadership and the group context. Researchers and practitioners can also use this approach to guide the development of the types of questions that provide insight into a group’s context. For example, what are the group’s current challenges and goals? What values and norms do the group’s identity consist of? How does the leader bring these values to life? And how do the followers perceive their leader(s)? As things stand, though, researchers are generally reluctant to answer such questions by examining effective leadership within the specific context where it occurs. So, by addressing this gap, they will be better placed to provide leaders with an understanding of the process through which leadership occurs. In this endeavour, we suggest that ethnographic methods—where researchers immerse themselves in leadership environments for an extended period of time—are likely to be particularly useful.

## 3. Ethnography to Understand Leadership as an Insider

Ethnography involves researchers immersing themselves in a particular group or culture to generate theory and knowledge through direct experience over time [27,28]. An ethnographic approach allows those researchers to gain firsthand experience through direct and sustained social contact with those in the group [29]. Not least, this is because by interacting, observing, and participating within the group, they gain contextual knowledge of the meaning behind different interactions, behaviours, and events.

Ethnographers often adopt a constructivist approach to research, which assumes that reality is socially constructed and differs between groups. A constructivist perspective is helpful for understanding leadership, as it acknowledges that there is no universal truth about what ‘good’ leadership is. Rather, this perspective acknowledges that how people experience a certain type of leadership will differ across people and groups.

Researchers who immerse themselves in the groups they examine can observe how the group is able to shape group members’ reality [27]. As described by the social identity approach, how we view ourselves and the values, beliefs, goals, and norms that we adopt are shaped in important ways by the groups we belong to and identify with [20]. For example, a player in a local football team may value friendship, playfulness, and teamwork, whereas that same player in a state representative side may adopt group values of individual success and discipline in the hope of being noticed by professional team scouts. So, in their local club, that player will likely be motivated to joke around with their teammates, whereas in their state representative side, they will likely be motivated to showcase high-level physical skills. In this way, the player’s values and behaviour are shaped by the different socio-cultural forces around them. Researchers can better understand a group’s particular socio-cultural forces by immersing themselves in a group environment for a prolonged period.

Ethnographic research has been used for decades to understand a group’s cultural world firsthand and to gather contextual insights that would otherwise be unattainable. An example is Lewin and colleagues’ [30] classic social psychology study that examined how different leadership styles affect group dynamics among children. The researchers set up three different groups, each with either a democratic, autocratic, or laissez-faire leader. The researchers then observed these environments over several months and noticed that in the autocratic group, children were aggressive, hostile, and uncooperative, and would only work when the leader was present. In the democratic group, children were more cooperative and took the initiative in tasks such as cleaning up and working even when the leader was absent. In the laissez-faire group, the children were uncooperative, tended to focus on their individual tasks, and became more easily bored.

Lewin and colleagues’ [30] research affords insight into the richness of context-specific data that are only accessible by directly observing a group. However, researchers do not need to manipulate that environment in order to do this. This is shown by another classic study in which Festinger and colleagues [31] embedded themselves in a religious cult whose leader, Mrs Keech, predicted an imminent apocalypse that would bring about the end of the world. The researchers acted as genuine believers to examine how people came to believe in such extreme ideas and how they dealt with their disconfirmation once the predicted end of the world did not eventuate. In particular, they captured the emotions of group members and the process through which these extreme beliefs and social bonds were actually strengthened once the prophecy had failed to materialise. These findings contributed to the literature on cognitive dissonance and social dynamics in a unique and powerful way. Unfortunately, though, these types of immersive projects have largely disappeared from contemporary science—including the leadership literature—as researchers have come to prioritise what are seen as more generalisable quantitative methods. Yet as Alvesson [14] argues, for researchers to better understand leadership, we need to examine leadership in its natural habitat—the ordinary and extraordinary environments like Mrs Keech’s cult where it really occurs.

There are nevertheless some notable exceptions that have used ethnographic methods to study leadership more recently. Over three years, Smith and colleagues [32] immersed themselves in an inter-organisational research and development team. They observed that an individual’s leadership status was shaped by their ability to make meaningful contributions to the group rather than their role or title within the organisation. Accordingly, it was often the informal leaders, not the formal leaders, who influenced their team. However, an employee’s capacity to influence their team was quickly undermined if they put their personal interests above the group’s interests. For example, group members often sought advice from a colleague who had specialised expertise in their area. By willingly sharing their expertise, group members listened to and followed the directions of this colleague. However, at a later point in time, this same colleague was perceived to prioritise the completion of their PhD thesis above the group’s projects, and so they gradually lost influence among the group. By immersing themselves in this organisation, the researchers could witness the leadership that happened in the corridors and common areas of an organisation, not just the directives of executives and managers in company meetings. The research shows how ethnographic methods can provide the contextual depth necessary to better understand leadership and offers a valuable approach for future leadership research and practice.

Alvesson and Jonsson [33] took a case study approach to examine a middle manager’s understanding of leadership in an international manufacturing company. They used interviews, reflections, and observations to observe that how managers described their leadership often differed from how they actually practised leadership. For example, a manager said that they strived to lead like a coach by listening to their employees and providing further questions to support employees’ problem-solving, instead of providing direct answers that discourage employees from thinking for themselves. However, the researchers observed contradictory behaviours from the manager, who was often passive during meetings, offered little support, and was sometimes critical and controlling of their employees. This is a good example of how observational data can provide unique insights that would not have otherwise been captured.

In a sporting context, Lara-Bercial and McKenna [34] used ethnographic methods to examine youth development in basketball. They noticed how many forces, such as the personal characteristics of players, interpersonal relationships, daily routines, and club culture, influenced player development in different ways. In particular, the researchers highlighted the fact that the club context was constantly changing so that each young player experienced the club environment differently. For example, on one day, a young player might feel connected and motivated to train, whereas on another day, after a dispute between teammates, that same teammate might feel uncomfortable and disconnected from their teammates. If the researchers had not been immersed in this team environment, it is unlikely that they would have noticed the ebbs and flows of the young player’s day-to-day experiences.

In another ethnographic study, Rothwell and colleagues [34] observed how a culture of masculinity and disciplined behaviour in English rugby academies contributed to rigid, authoritarian coaching styles that shaped players’ thoughts, feelings, and behaviours. The researchers noticed that these coaching styles were deeply embedded in the culture, which limited player autonomy and creativity. However, because researchers immersed themselves in the environment, they observed that often before official training began, players initiated a range of rugby-like games in which the athletes played with creativity and openness, unlike their official training. It was the context of the coaches being present or not that limited the players’ freedom to be creative and open to new ideas.

In all of these examples, because the researchers immersed themselves in group environments, they noticed how peoples’ behaviour was shaped by the group context at that point in time. If the researchers had not immersed themselves in these environments and instead acted like external researchers who observed and/or surveyed the team members at one or two timepoints, then it is unlikely they would have appreciated the detailed behavioural changes and salient contextual factors (e.g., team conflicts, manager behaviour, and coach presence) that contributed to those behavioural changes. In these cases, as researchers immersed themselves in these environments, they came to live, feel, and experience the group environment in a way that provided contextual data beyond the scope of surveys and one-shot interviews.

### 3.1. The Role of the Ethnographer

Immersion in the field can vary from the researcher being a full participant to being an observer, or somewhere in between [35]. Ethnography can be theory-driven yet dynamic and adaptable, responding to environmental demands. There is no one-size-fits-all approach to ethnographic research, and this allows researchers to adapt their methods to the specific situation and context. In some cases, the researcher may be an outsider with minimal or no prior experience within the group or culture. Alternatively, the researcher could be an insider and a pre-existing member of the group or culture. For example, a professional boxer could examine the boxing culture while still competing [36]. In this way, professionals like Champ and colleagues [37] have leveraged their pre-existing group membership to gain an inside perspective on the experience of health professionals in a sporting environment.

A researcher’s involvement can range along a continuum from observer to participant [35]. As an observer, the researcher stands back and observes the group in action. As a participant, the researcher engages in some or all group activities. For instance, a researcher might participate in some of the training drills with the group. This type of involvement can be flexible, and researchers can adapt their approach depending on what seems most appropriate. Indeed, as researchers spend more time in the field, they often transition from being outsiders to insiders and come to feel more comfortable acting as participants. Participating in group activities can help build connections within the group while also allowing the researcher to observe and experience interactions that might go unnoticed when observing from a distance (e.g., making sense of brief comments and interactions among players during training drills). By participating in group activities, researchers can also assist the group (e.g., helping out during specific training drills) in ways that allow the researcher to contribute to the group and strengthen their rapport with group members. This speaks to the fact that the relationship between the researcher and the group should be reciprocal and mutually beneficial to both parties.

### 3.2. Methodological Considerations

Researchers should clearly define their study focus by deciding in advance on the theoretical lens(es) through which they will interpret their data [38]. This process helps researchers narrow their observations and analyses, making it easier to distinguish relevant from irrelevant information. Ethnographers often use multiple data collection methods in the field, interconnected with their participant observation. Initially, these observations may be broad, before gradually narrowing down to focus on specific aspects of group life. For example, by first noting general trends and behaviours within a group, a researcher might notice changes in typical behaviour and subsequently focus on discerning these changes. Extended observation allows researchers to experience the ebbs and flows of a season first-hand (often influenced by performance outcomes such as victory and defeat).

Researchers may also engage in informal conversations and interviews with group members. In leadership research, informal conversations with followers can yield rich insights into the ways that the group experiences different leadership behaviours. For instance, a researcher might ask, “What did you think of that?”, prompting the respondent to describe their experience. The researcher can then probe further and clarify specific details. Over time, as the researcher becomes more embedded within the group and develops rapport, group members might initiate conversations, providing more opportunities for open discussions about their interests or concerns.

Interviews, whether structured or semi-structured, are another valuable data collection method. These interviews can help gather in-depth information about the significance of particular events within the group. By being immersed in an environment, researchers can observe these events and then ask clarifying and probing questions in the interviews. Moreover, by sharing experiences with group members, researchers can ascertain and understand the cultural language that is being used (e.g., the significance of particular personal references or in-jokes). This language can help them connect with interviewees and conduct more insightful interviews. The flexibility of semi-structured interviews also allows the researcher to ask unplanned questions that probe further exploration [39]. Interviews can often be interwoven with participant observation to expand on the researcher’s observations and lived experience [40]. The combination of observation and interview data can be particularly beneficial for understanding group members’ perspectives after the researcher has observed specific leadership interactions.

Kelly [41] suggests that when researchers observe leadership behaviours, they are in a position to consider not only the salient, ‘grand’ actions of leaders (e.g., giving speeches and leading meetings), but also more subtle actions (e.g., informal conversations, words of encouragement, and expressions of care). The social identity approach recognises these subtle actions as an important aspect of the leadership process. For example, even though it may not be as widely heralded as giving an inspiring speech, expressing care and assisting a group member who might be struggling with their work or personal life can be a powerful act of leadership. By expressing care and assisting that group member, a leader could strengthen a shared social identity by demonstrating that they are concerned with a person’s membership in the group and that they are in this together. These more subtle leadership behaviours can help leaders build their capacity to influence fellow group members, because those group members are more likely to perceive them as one of ‘us’ who is leading for ‘us’. Researchers and practitioners who adopt ethnographic methods are much more likely to have access to these more subtle leadership acts from both informal and formal leaders.

### 3.3. Challenges of Ethnography

Ethnography is inherently complex, as the ethnographer’s presence influences both the events that are observed and the relationships formed in the field, and these in turn contribute to the ways that the researcher understands and describes the culture [42]. The researcher cannot remain neutral in the process of explaining and representing cultural and social life or detach themselves from analysis and representation; instead, researchers are always implicated in the ways that their cultural understandings are portrayed [43]. This issue of researcher subjectivity is an uncomfortable yet unavoidable reality of the ethnographic process [42]. However, there are moments when ethnographers can leverage their subjectivity to interpret the culture more deeply [44]. Ethnography involves more than merely participating and observing; it requires the ethnographer to immerse themselves in the lived experience of the group, feeling and experiencing from first- and third-person perspectives. This delicate balance of being both an insider and an observer over an extended period is a hallmark of ethnographic research.

To address the subjective biases inherent in ethnography, researchers are encouraged to engage in reflexive practices. Despite the recognised importance of reflexivity in qualitative research, there is a lack of empirical and theoretical work clearly outlining the process. Indeed, there is considerable conceptual ambiguity associated with the term reflexivity itself [45,46]. Nevertheless, reflexivity demands that researchers critically examine their own positioning within the field, considering how their pre-existing experiences, current situation, and research practices influence their analysis [47]. A detailed discussion of reflexive practices is beyond the scope of this article, but is the focus of a number of excellent treatments elsewhere (e.g., Townsend and Cushion [48]; Wetherell and colleagues [49]). Ultimately, though, reflexivity is a process that requires researchers to take time to view and consider the ways in which their own perspectives and identities shape their understanding and representation of the cultures that they study.

### 3.4. Presenting Ethnographic Research

Ethnographic data can be developed into a thematic narrative, which tells a story constructed from the researchers’ observations and analysis [50]. The narrative is built on a main idea that often includes several analytic themes explored throughout the narrative and later discussed in the study [51]. As part of this process, Wolcott [52] suggests that ethnographers should begin with a description of the culture that describes “what is going on here?” (p. 12). The data are then analysed by identifying patterns, contextualising information within a broader analytic framework, and connecting them with the personal experiences of both the researchers and the groups under investigation. Creswell and Poth [51] recommend that findings be both discussed on the basis of the researcher’s personal experiences and connected to broader scholarly research on that topic.

To present these results, ethnographers can take on the role of storyteller by developing examples, cases, and vignettes to illustrate the key themes. Here, ethnographers create scenes that engage the readers’ minds in a way that positions them in the characters’ shoes to experience what it might feel like to be part of that group [53]. All in all, ethnographers “tell a good story” [54]. Specific to leadership research, we see an opportunity for ethnographers to write stories about leadership that illustrate leadership in context. These stories are a way of showing readers what effective and ineffective leadership look and feel like rather than simply listing what effective leadership is. So, by telling a leadership story, readers are immersed in a world where they see and feel how leadership occurs in context.

Researchers can use creative non-fiction to produce stories to illustrate their research findings [55]. Creative non-fiction in academia gained traction as scholars questioned the prevailing norms in human sciences, particularly in regard to the ways social reality is described and how participants’ lived experiences might be captured effectively [53,56]. This shift in how we meaningfully communicate research in leadership has led researchers to explore alternative ways to present research, with an emphasis on giving voice to participants and describing the context in which they think, feel, and act [57]. In this regard, writing in the social sciences has been criticised as ‘unpopulated’, such that research omits the reality of the subjects the research was about. Instead, researchers write about ‘fictional things’, which can be vague and confusing [58]. A creative non-fiction approach can steer researchers away from this type of writing and bring life back to subjects by describing their story and fleshing out the context in which they find themselves.

The approach that is required to produce creative non-fiction stories fits well with the process of ethnography. For instance, Gutkind [59] explains that creative non-fiction stories require writing to have real-life immersion, reflection, research, and reading and writing—all of which are key aspects for completing ethnographic research. In the field of sport and exercise, Sparkes [56,60] was a pioneer of this method as a way of challenging traditional modes of research representation. He emphasised the need for different writing styles and a broader representation of participant voices. More recently, the application of creative non-fiction in sport and exercise research has been expanding, as noted by Cavellerio [53]. However, our own experience suggests that researchers are often hesitant and uncomfortable exploring new boundaries and expanding beyond orthodox methods when it comes to communicating their research.

Despite this discomfort, we encourage researchers to challenge themselves to explore creative non-fiction. Humans are storytelling animals, naturally drawn to stories [61]. By bringing research to life through stories, researchers can provide the reader with a rich contextual background of the key themes being explored. Furthermore, stories provide a way of sharing research findings in a language that reaches beyond the academic community [55]. It provides an opportunity for researchers to transform their findings into compelling narratives, which are shared between and appealing to leaders and followers themselves, not just fellow academics [53].

## 4. Integrating Ethnographic Findings into Leadership Development Programs

In choosing to undertake an ethnographic study, researchers are fully aware that their findings are not generalisable to other contexts. Generalising findings to other settings and contexts is not the aim of ethnographic research. Instead, the key characteristic of ethnographic research is its specificity to a particular group and/or culture. This is both its strength and limitation. For while the inability to generalise findings can be seen to limit the applicability of conclusions, the payoff is a comprehensive understanding of a specific context. Moreover, as the quality and quantity of ethnographic case studies grows, researchers and leadership educators will be better positioned to provide a diverse set of examples of different leadership styles in various contexts.

One promising use of leadership case studies in leadership development is to stimulate discussion and problem-solving among leaders. For instance, it is possible to consider incorporating creative non-fiction stories into leadership development programs. Leaders could be presented with a specific issue or challenge, provided with a leadership story, and asked to discuss the strengths and weaknesses of the leadership observed in the story, along with their recommendations. This type of problem-solving engages leaders in higher-order thinking, enhancing their learning experience and perhaps subsequently affirming and/or re-shaping their thinking and behaviour. Furthermore, if conducted within a particular group, this activity could engage fellow leaders in collectively solving leadership challenges within their group, and help them to better understand where their fellow group members are coming from.

In the context of leadership development, the primary aim of ethnographic research is to support and challenge pre-existing research and leadership development programs, by providing detailed and rich accounts of how certain leadership behaviours can be effective and ineffective in a specific situation and context. By incorporating multiple case studies into leadership development programs over time and across settings, leadership educators can provide examples of various leadership behaviours in different contexts. These can demonstrate how and why the effectiveness of different leader behaviours varies across these contexts. In this way, these programs can help leaders to discern which leadership behaviours and processes might be more suitable for the specific situations and the context in which they are operating, and to have a broader and more grounded understanding of the forms that effective leadership can take. As suggested above, by engaging leaders in discussions with these case studies, leadership educators can go beyond describing the ‘what’ of leadership, to explaining the ‘how and “why’ of leadership.

In these various ways, an ethnographic approach has the capacity not only to deepen researchers’ understanding of leadership but also to support the work of practitioners and leadership consultants working in the field. By immersing themselves in group environments with the specific aim of observing leadership behaviour, practitioners can conduct an audit of how leadership is enacted within a particular group and identify the group’s specific needs. These leadership insights can then provide a foundation to advise formal leaders on how to adopt more effective leadership practices and embed these more widely in the team. Throughout this immersive process, practitioners could engage in an iterative cycle of information exchange with formal leaders, continually sharing their insights to collaboratively shape their understanding of leadership in this group. And in doing this, an ethnographic approach ensures leadership development and advice are always grounded in the specific context of the group.

A thorough understanding of the team’s context can also help researchers and practitioners to provide tailored recommendations for a given group. As part of a group’s leadership development and education, practitioners could share their leadership observations with the entire group. Doing so could help to develop a group with more inclusive and shared forms of leadership. By highlighting the influence of informal leaders, practitioners can reinforce and encourage further leadership throughout the group. This feedback to informal leaders may reveal an unrealised understanding that informal leaders exhibit leadership qualities and behaviours that benefit their group. Feedback shared with the entire group can broaden the group’s perception of leadership beyond formal titles and positions of power, and instead encourage the idea that all group members have the potential and agency to lead and influence others within the group.

In sum, by shedding light on group members’ lived experiences, practitioners and leadership consultants who adopt an ethnographic approach can help groups and formal leaders understand how leadership outcomes are often shaped by followers, the group, and the broader context. By embedding themselves within a group and sharing these leadership insights, researchers and practitioners can guide the group with a tailored roadmap for improving their leadership.

## 5. Conclusions

The present paper encourages researchers and practitioners to adopt methods that examine the dynamics between leaders, followers, and the group context to better understand leadership and associated outcomes. Researchers who use ethnographic methods can capture some of the complexity, dynamism, and fluidity of leadership in action. In line with these suggestions, Figure 1 outlines a process that researchers and practitioners can follow when using ethnographic methods to examine leadership.

First, researchers and practitioners should consider the strengths and limitations of ethnographic methods and how they could best support their leadership research. Next, they should consider an appropriate setting for their research and reflect on how they might engage that group. Here, once a researcher is engaged with a group, they could follow a continuous cycle while collecting data. Researchers should be clear about their role within the team, but be conscious of the capacity (and need) to adapt this in the course of the immersion. For example, a researcher might initially take a more passive role as they observe the group dynamics. Then, as they gradually build rapport with the group, they might take on more active roles and participate in team activities.

Participating in group activities also allows researchers to experience leadership within the group. As part of developing rapport with the group, researchers could be open and transparent about their role and observations. Doing so can help them build trust among the group. While sharing insights with group members, researchers can initiate discussions with group members that provide insights and suggestions for the researcher’s interpretations. The researcher can also engage in an iterative process with the formal leader(s) in which they share formal and informal feedback, and together, the leader and the researcher can make sense of the data. Indeed, by involving formal leaders in the data collection and analysis process, researchers are giving back to the group, building rapport, and collecting insightful data. Feedback can also be an ongoing process where researchers provide progress reports to the group or to particular members (e.g., formal leaders). Doing so can support the sense-making process and also demonstrate transparency in the research process.

Throughout this entire process, the researcher should also consider how they can work with and within the group in an unobstructive and unobtrusive way. There is no one way to do this, and to a large extent, this requires social skill on the part of the researcher to discern how they might best navigate their way through the research process and their membership in the group. At the conclusion of the project, researchers can share a final report with the group or its formal leader(s) and discuss their conclusions. Doing so ensures that the group and the researcher benefit from the research, and allows the researcher to clarify their interpretation with the group and its leaders. It may benefit the entire group to share feedback and a final report; however, the researcher should consider what forms would be most beneficial and consult the formal leader(s) about the possibility and utility of sharing these insights with the broader group.

In sum, by embedding themselves within groups, ethnographers can create meaningful stories and narratives that illustrate how leadership outcomes are shaped by the interplay between leaders, followers, shared identity, and group context. These leadership narratives can support leadership development practices by providing context-specific examples of leadership in action. By including these examples in leadership development practices, leaders can observe how leadership effectiveness is intertwined with their relationship to followers and the context. Furthermore, these leadership narratives have the capacity to engage leaders in problem-solving activities that are based on real-world scenarios that can enhance a leader’s ability to form and adapt their leadership approach to meet the demands of the context. Through this approach, leadership development can incorporate an interactive and reflective process that prepares leaders to navigate the complexities of their environment and that helps them and those who support them to create more leaderful teams.

## Figures and Tables

**Figure 1 behavsci-14-00946-f001:**
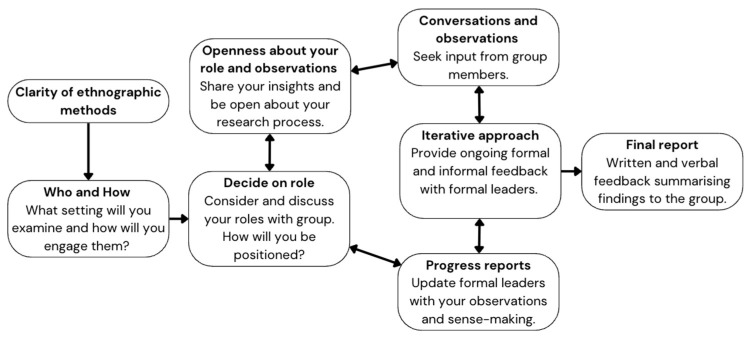
Ethnographic process to examine leadership.

## Data Availability

Not applicable.

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
