# Peer review of "Understanding Leadership from the Inside: Using Ethnographic Methods to Examine How the Interplay between Leaders, Followers, and Group Context Shapes Leadership Outcomes"

_behavsci, 2024, doi:10.3390/bs14100946_

Round 1
Reviewer 1 Report
Comments and Suggestions for Authors
An excellent and inspiring article on ethnographic research on leadership. Researchers, scholars, students and also practising leaders can benefit a great deal from the article.

Reviewer 2 Report
Comments and Suggestions for Authors
In general, this is a well written paper that canvasses relevant literature. However, it reads as if it arises from a literature review for a PhD on leadership (possibly in sport), that also encompasses literature related to the methodology. This leads to my major reservation that the paper does not really extend our understanding of leadership or ethnography (even noting that it is an opinion paper)
Thus, around the criteria of novelty, significance and merit I would not recommend publication.
If the paper were to be reshaped for publication, I would raise the following issues for consideration:
· It is an odd mix of leadership literature from what might be called organisational leadership and sport leadership/coaching. If the underlying aim of the paper is about setting up a study for sport leadership/ coaching, this should be explicitly stated at the commencement, and subsequent discussion reorganised to reflect this reality. Alternatively, if the paper is trying to draw together a range of leadership literature, it could explore possible tensions, differences and confluences between these bodies of literature.
· Within the organisational leadership literature, the leaders/ followers typology (to me), sets up a false dichotomy about who is who. It also implicitly denies the agency of those that are not positional leaders. But perhaps it makes better sense in a sports context?
· Although the authors cite another source for this idea, I am uneasy with leadership being defined as a ‘wicked problem’. A wicked problem is usually applied in the public policy space in relation to seemingly intractable and complex problems. Sure, leadership might be complex and require context specific approaches, but this does not push it into the space of a ‘wicked problem’. There are competent and excellent leaders to be found in many places, which in itself undermines this description.
· I like the suggested strategy of examining leadership through ethnography, and ethnographers using stories to illustrate leadership in context. However, it would be important to be clear about whether the observations and stories concern ‘leadership’ (therefore, how will you identify acts of leadership) or those who happen to be in positions of leadership. The latter focuses on the person in positional power, who may or may not be acting in a leaderful way.
In short, in its present form, the paper does not extend our understanding of leadership. The suggested ethnographic approach is interesting, but I suspect the paper arising from the research once it is conducted will be much richer.
Reviewer 3 Report
Comments and Suggestions for Authors
A great conceptual idea presenting ethnography as a tool to further explore how leadership processes occurs. Thank you for presenting a detailed explanation of a very complex concept that it is the adoption of this method. Authors have clearly defined it and have been able to provide an excellent explanation of ethnography including the counter argument of biases. I congratulate them on this. I only have two suggestions for improving your argument.
1. It is recommended to strengthen the argument on how been part of a team environment can provide an in depth understanding of leadership. As it reads, even with correct citations, it reads as an assumption. This can be improve by providing an additional example in the context of business/management scenario. A much more realistic scenario would then provide a substantial value of the use of ethnographic methods.
2. in the concluding section, the paper could benefit by expanding a description (or an info flowchart) of the steps that one can undertake if decide to adopt this approach. Including summarising the challenges that can be overcome by the value of an ethnographic study.
overall , I enjoyed this opinion paper , however I felt the example used throughout the paper is too simplistic to demonstrate and validate the propose argument. If an additional business real example can be provided, it would be useful.
Round 2
Reviewer 2 Report
Comments and Suggestions for Authors
I appreciate the care taken in addressing the previous concerns raised. I note that you have submitted a revised manuscript with comments on the side - these will need to be removed.
Author Response
Comment 1: I appreciate the care taken in addressing the previous concerns raised. I note that you have submitted a revised manuscript with comments on the side - these will need to be removed.
Response 1: Thank you for pointing this out, these have been removed.